# An Individual’s Lived Experiences of Taking Cannabis-Based Medicinal Products (CBMPs) to Treat Anxiety

**DOI:** 10.3390/ijerph20186776

**Published:** 2023-09-18

**Authors:** Samantha Hurn

**Affiliations:** Department of Social and Political Sciences, Philosophy and Anthropology, St Luke’s Campus, University of Exeter, Heavitree Road, Exeter EX1 2LU, UK; s.hurn@exeter.ac.uk

**Keywords:** CBMPs, anxiety, lived experience, N = 1 trial, autoethnography, medical anthropology

## Abstract

This report documents the case of a patient (the author) participating in a clinical trial of medical cannabis (*Cannabis sativa* L.)—the Sapphire Access Scheme, run by the Sapphire Medical Clinic as part of the UK Medical Cannabis Registry—to explore the impacts of cannabis-based medicinal products (CBMPs) on anxiety. For most of my life, I have experienced often very serious bouts of poor mental health arising, in part, from childhood abuse, and have been diagnosed with several mental health conditions which constitute disabilities. I have received various conventional treatments and multiple alternative therapies. However, none of these have enabled me to consistently manage my conditions long-term, and I often suffer relapses. As part of the Sapphire Access Scheme, I complete regular quantitative questionnaires regarding the impacts of the CBMPs on my anxiety and have also obtained the clinic’s permission to qualitatively document and write up the impacts of CBMPs on my mental health. Here, I present a preliminary autoethnographic exploration of my lived experiences of CBMP use over the first four months of the trial, which show that even within such a short space of time, CBMPs have had a positive impact on treating what had previously been treatment-refractive chronic anxiety.

## 1. Introduction

Since 2018, it has been legal in the UK to prescribe medical cannabis (*Cannabis sativa* L.) or cannabis-based medicinal products (CBMPs) for a limited spectrum of physical and psychological conditions. While there is also growing evidence for the benefits of the off-license use of CBMPs for treating other neurological conditions including anxiety as well as chronic pain, “a critical barrier to access is a lack of the right evidence to enable the NHS to fund medical cannabis” [1]. The UK Medical Cannabis Registry currently has 5000 patients contributing quantitative data regarding the effects (both benefits and adverse events) of medical cannabis on their general health and quality of life. To date, there have been a limited number of qualitative studies focused on the experiences of medical cannabis users, and these have all flagged the need for further qualitative as well as clinical research.

Ryan and Sharts-Hopko [2] reviewed extant qualitative research on the experiences of medical cannabis users in Europe and the USA, and stigma and risk emerged as the primary themes. Stigma related both to how prescribed users felt others perceived their medical cannabis use (and concerns of being associated with stereotypical representations of recreational cannabis users), as well as concerns for how medical professionals would perceive their use of medical cannabis (where there was a lack of knowledge about the potential benefits of cannabis and a disproportionate emphasis on potential risks). Risk was expressed in relation to the illicit status of the drug in some countries or states, as well as weighing risks of medical cannabis use against the known side effects and benefits of mainstream prescription drugs. Risk also featured in the negative outcomes that some users experienced, such as adverse effects on cognition or balance when the dose was too high, which also tied back into the stigma associated with recreational cannabis use and stereotypes. Hulaihel et al. [3] conducted interviews with a sample of 15 medical cannabis users in Israel who also felt social stigma as a result of taking medical cannabis which negatively impacted their sense and presentation of self. Schlag et al. [4] conducted interviews with 11 families in the UK with children who had been prescribed medical cannabis for epilepsy. The findings from this specific cohort foregrounded frustrations regarding national policy and the regulation of medical cannabis, and challenges experienced due to the lack of knowledge and understanding of medical cannabis within the wider UK healthcare system.

More recently, Garcia-Romeu et al. [5] conducted online surveys with a large sample (N = 808) of medical cannabis users who took medical cannabis for either neurological issues or chronic pain. The majority of their survey participants reported that medical cannabis had provided relief for conditions where conventional treatments had failed. However, concerns were also raised along similar lines as previous research, primarily in relation to risks (side effects, legality) and stigma (especially in relation to a lack of wider knowledge and the risk of misinformation). Financial implications were also flagged as a concern, and a potential obstacle to use. Garcia-Romeu et al. concluded that their data indicated that “greater research and education on the safety and efficacy of medicinal cannabis/cannabinoid use is warranted.” This echoes the recommendations of Banerjee et al. [6] following an analysis of the current data from the Medical Cannabis Registry. They stated that “Novel approaches of data collection and analysis will be integral to improving clinical evidence on CBMPs. [Real world evidence] RWE can be used in conjunction or as an extension to [randomized controlled trials] RCTs to increase the speed of evidence generation, as well as reduce costs.”

In light of the limited extant qualitative data, the current lack of any systematic accounting for the day-to-day lived experiences of individuals taking CBMPs, as well as the recognised need for further research that incorporates real-world evidence, I present here a case report based on an individual patient’s (the author’s) lived experience of the impacts of CBMPs on anxiety over the first 4 months of treatment. While this may appear to be too short a time frame for CBMPs to have had any appreciable impact, there is a precedent amongst researchers also documenting positive impacts in a shorter window. Bar-Lev Schleider et al. [7], for example, found that over half of the children and young people with Autism Spectrum Disorder (ASD) in their sample who took CBMPs experienced improvements in their symptoms, with 31.1% reporting moderate improvements and 48.7% reporting significant improvements after just one month on the trial.

## 2. Detailed Case Report

For most of my life, I have experienced often very serious bouts of poor mental health. I have survived significant and prolonged childhood abuse, as well as a series of other traumatic experiences, including, for example, coma and lengthy hospitalisation with bacterial meningitis, being violently abducted at knife-point, and raped as a teen. I have been diagnosed with several mental health conditions linked to childhood abuse and trauma including depression, anxiety, post-traumatic stress disorder (PTSD), obsessive compulsive disorder (OCD) and attention deficit disorder (ADD), which at times can negatively impact my ability to function to the extent that they constitute disabilities. In November 2022, I was accepted onto a clinical trial of medical cannabis—the Sapphire Access Scheme, run by the Sapphire Medical Clinic as part of the UK Medical Cannabis Registry—to explore the impacts of CBMPs on anxiety.

As the Sapphire Research Director, Simon Erridge notes, “[CBMPs] can only be started by specialist doctors if alternative treatments have been tried without providing adequate symptom relief” [1]. In my case, while I suffered with anxiety from an early age, I only started receiving extensive and varied conventional treatments after experiencing a period of intense emotional distress (which at the time was referred to as a ‘nervous breakdown’) at 14 years of age, when my OCD and anxiety became so debilitating that I was unable to leave the house. Treatments have included multiple courses of talking therapy (including cognitive behavioural therapy and emotional management), pharmaceutical anti-depressants and EMDR (eye movement desensitisation and reprocessing therapy), as well as alternative therapies including acupuncture, hypnosis, herbal medicines and physical interventions such as tailored exercise regimes. However, none of these treatments have enabled me to successfully or rather consistently manage my conditions long-term, and I often suffer relapses. At the time of joining the trial, I was not receiving any other form of treatment.

As part of the Sapphire Access Scheme, I complete regular questionnaires regarding the impacts of the CBMPs on my anxiety. Given my unusual position as a social scientist who often adopts an autoethnographic approach (drawing on my own lived experiences) to inform the research that I undertake in a professional academic capacity, I have also obtained the Sapphire Clinic’s permission, and the approval of the ethics committee within my own institution, to qualitatively document and write up the impacts of CBMPs on my mental health over the course of a year. I am conducting an autoethnographic exploration of my experiences, much along the lines of an N = 1 medical trial. Here, the focus is on observing and documenting changes in individual cases over time, which, it is argued, can provide important evidence to enhance individualised care and treatment plans [8,9].

In 2000, during the fieldwork for my PhD, I started to keep a detailed reflexive diary which initially took the form of a handwritten document in a series of physical paper notebooks, and then migrated to the notes feature of an iPhone after I first acquired one in 2014. I typically jot down thoughts, experiences, observations and anything else which strikes me as being important or interesting as soon after the event as possible. Since joining the Sapphire Access Scheme, I have tried to systematically detail key aspects of my day and a summary of my mental health last thing in the evening before going to bed. I also make a note of what I consider to be other key variables, such as what I have eaten and drunk, what exercise I have done, the quality and duration of my sleep and so on, as well as the dose of CBMPs taken, which I adjust according to the particular circumstances or challenges of each day. For example, if I know I have something stressful to deal with the next day, I will increase the dose on the preceding evening, or if I have had a relatively stress-free few days I will reduce it. However, this has been a case of trial and error. The maximum dose I have taken on any single day is 0.5 mL of THC and 0.8 mL of CBD (with the latter split into two doses: one of 0.3 mL taken during the day, and another dose of 0.5 mL taken in the evening before bed in combination with the single 0.5 mL dose of THC). My average daily dose across the trial to date has been 0.2 mL of each substance.

I am using diary entries as my primary data source to create the autoethnography. As noted by Adams and Herrmann [10] in their introduction to the first issue of the *Journal of Autoethnography*, “autoethnographic projects use selfhood, subjectivity, and personal experience (‘auto’) to describe, interpret, and represent (‘graphy’) beliefs, practices, and identities of a group or culture (‘ethno’).” Personal diary entries can be used as mnemonic aids to facilitate reflections, as well as data sources in their own right, and these will assist me as I attempt to “describe, interpret, and represent” what it means for me to experience anxiety and depression, prior and subsequent to starting the trial. This can then potentially inform how others with similar backgrounds and symptoms might engage with CBMPs.

Autoethnography is particularly useful for research that seeks to explore the lived experience of mental health and the lived experience of medical cannabis treatment because it provides a means of “figuring out what to do, how to live, and the meaning of [one’s] struggles” [11]. As Bochner notes, “Autoethnographers are in the business of storying lives. As storytellers, we are preoccupied with plots, and plots are driven by misadventures, reversals of fortune, blows of fate and lives spinning out of control” [12]. My medical cannabis journey started as the story of a life spinning out of control due, in part, to misadventures and blows of fate, but as will be revealed, the CBMPs have catalysed a reversal of fortune, enabling me to regain some control over my symptoms and therefore my life.

Autoethnography also facilitates critical reflection on the ways that the researcher’s experiences intersect with and impact on the wider context of the family [13]. Indeed, Little and Little have noted that autoethnography “frequently focuses on some aspect of vulnerability […], but simultaneously links our lives to those that we live with” [14]. Chang reminds autoethnographers to carefully consider the ethical implications of this methodological approach. She states that “As you play a multi-faceted role as researcher, informant, and author, you should be reminded that your story is never made in a vacuum and others are always visible or invisible participants in your story” [15]. Writing about my experiences is not possible without brief mention of my family, because my disabling anxiety is something that they too have had to live with, and the benefits that I have experienced since taking CBMPs have also indirectly been enormously beneficial for them. However, in focussing only on my own experiences and keeping my significant others present but in the background, I am emulating an established precedent from within the field of ‘family studies’, epitomised by the contributions in Wyatt and Adams’ [13] edited collection, where researchers reflected on the intersections between their own and family members’ experiences while foregrounding the former to protect the latter.

In this case report, I will provide a summary of some of the main impacts of the CBMPs on my anxiety over the first four months of the trial. For context, I will begin with an overview of my general mental health pre-CBMPs before comparing my experiences of medical cannabis use with those of other patients with similar or comparable backgrounds.

One of my earliest childhood memories is of standing on my maternal grandparents’ sofa, reaching out towards one of the many porcelain ornaments on the window sill. My Grandfather hit me so hard I not only fell off the sofa but was propelled across the room where I crashed into the centrally located coffee table. This is the first I can recall of many such violent encounters. Then, there are other memories of very different interactions with him, but these I have spent a lifetime trying to forget. It was not until I discovered I was pregnant in adulthood, long after the perpetrator, my maternal Grandfather, had died, that the decades of sexual and emotional abuse I had experienced in addition to the violence described above catalysed a relapse into heightened OCD and extreme anxiety. How would I keep my child safe in a world where I had experienced so much trauma? I underwent a lengthy process of referral and assessment as the local mental health services attempted to place me, and on her first birthday I eventually started weekly outpatient psychotherapy at the hospital where my daughter had been born.

I have always found talking therapies to be emotionally and physically draining, and while I was able to slowly come to terms with what had happened to me, and develop strategies to help me manage my symptoms, I found it almost impossible to implement what I had learnt in my daily life. After multiple courses of talking therapies over my lifetime, I had come to suspect that my anxiety was resistant to treatment, something which has been termed ‘treatment-refractory’ anxiety [16], in the sense that conventional treatment has never resulted in any long-term relief for me.

Over the years, my anxiety has been triggered or exacerbated by various factors including work stress, living stress, and financial stress, as well as grief, compassion fatigue, and caregiver burden for the many rescued and companion animals who are part of my extended family. When my symptoms have been severe, I have felt as though my loved ones and dependents would be better off without me. At work, I have often existed in a state of hyperanxiety, lurching from one panic attack to the next, finding the commute overwhelming and struggling to be on campus, let alone be the social and productive academic and supervisor that my colleagues and students need me to be, despite also being a workaholic. Feelings of imposter syndrome and inadequacy were fed by the competitive and fast-paced nature of academia, as no matter how hard I worked (often late into the night and during weekends), I was never able to stay on top of my ‘to do’ list and inbox.

During the first year of the pandemic, after being made homeless for a second time due to the loss of my house with negative equity, and the precarity of the rental market which was exacerbated by the need to rent land for our large number of rescue animals, we were forced to move into unsafe accommodation. Work stress also became significant at this time. My feelings of failure and hopelessness hit an all-time low. I was receiving weekly talking therapy (consisting this time of a mindfulness-based emotion management course), which, as with previous experiences of such interventions, helped during and briefly after each session, but my anxiety levels quickly went back to their default. I then tried chemical antidepressants for the third time in my life, but as I had experienced in the past, my symptoms were exacerbated and suicidal and other intrusive thoughts became overwhelming and so I discontinued their use. Out of desperation, I applied to join the Sapphire Access Scheme to try and help with what had escalated into disabling anxiety.

The following is an excerpt from my diary which describes a typical weekday morning at this time, prior to starting treatment with CBMPs.

“We are getting into the car. I happen to glance down as I put my phone in the holder and an email flashes up on my lock screen. I curse myself for forgetting to turn off notifications as without thinking I scan the content and feel as though I have been winded. Simultaneously my vision clouds, and dark shadows swim across my vista while the edges grey and close in. The shadows, floaters, are always there and easily ignored, but now they demand and command my attention. The world beyond these ghostly shapes blurs into insignificance. My breathing becomes shallow, raspy, ragged. I feel as though I’m drowning in air. I can’t inhale. The weight of the world is crushing my chest and as I fight to suck in a breath I feel a surge of adrenalin to my extremities. The nerve endings in my hands and face prickle and go numb, my arms itch and I am drawn to scratch them obsessively, although the action creates further irritation. My right eye begins to twitch. My mind is fog. Disjointed thoughts drift just out of reach. I can’t make sense of them and that exacerbates the panic. Nausea sweeps through my body and I wretch. I start scratching at my face. Palms sweaty, body tingly, I fear I may lose consciousness. Then anger wells up and pushes the nausea aside. My field of vision narrows and intrusive thoughts force their way into my mind fog. I clench my fists, my heart races, I want to run and scream and hide. The nausea returns as the adrenalin rush ebbs. I am exhausted. As the fog clears I hear ringing in my ears. My eyes recoil from the light of the outside world and my head pulses with a dull rhythmic ache. My fingers, arms, and legs tingle then feel heavy. I’m not sure I can move. I try to focus on the tasks I need to complete. What was I doing before this attack? I just can’t remember. Then a small voice penetrates the tumultuous thoughts and a little hand takes hold of mine, grounding me like a lightening rod. ‘Mummy, are we going now?’ I have emerged from the time warp of crippling panic, and oxytocin floods my system as my love for my daughter prevails. But yet again we are late for school. Then the feelings of shame and self-loathing kick in. I try to pick myself up and be the attentive and present parent she deserves. I kiss and hug her and we talk and sing and hold hands on the drive as I try with all my might to mask the after effects of the attack and focus on her voice and the love I feel for her. This too is so powerful it also threatens to overwhelm. I try to be kind to myself, to remember and implement the grounding techniques I’ve learnt and practiced in so many therapy sessions, whilst wrestling with the overwhelming feeling that I fail her, that perhaps she would be better off without me, but that simultaneously I can’t bear being apart from her. I feel hungover, as the aftershock of the panic attack and my feelings of failure linger, making every subsequent task that day a battle…”

## 3. Discussion

As will be revealed below, it is no exaggeration to say that the CBMPs have changed my life and such disabling and terrifying episodes are no longer part of my daily existence. For the first time in memory, I feel in control of my mental health. Although in the process of documenting various other factors as part of the diarising entailed in autoethnography, I have also come to note the influence that workload and working patterns, diet, sleep, exercise and menstrual cycle, for example, can also exert, but the analysis of these factors falls beyond the scope of the current case report. Right from Day 1, however, within less than an hour of taking my first dose, comprising 0.1 mL of CBD (50 mg/mL) and 0.1 mL of THC (20 mg/mL) oils, I felt as though a massive weight had been lifted. While I have always been able to ‘mask’ my internal turmoil and hide it from the majority of people in the outside world, one of the things I have struggled with is finding the strength and space to ‘step back’ from any negative or triggering situation or feeling. My ‘window of tolerance’ has traditionally been narrow, with very little external input needed to push me beyond my ability to cope and into a highly reactive (in the sense of panicked) state. However, what has been the most significant positive impact of the CBMPs is the sense of being cocooned in a protective blanket, of frayed nerves being soothed by the cannabis’ presence in my system, of space being created between me and the triggers and threats of the outside world. It has enabled me to ‘do the work’ that the talking therapy advocated but which had previously been impossible to achieve in practice. I sometimes notice that the effects of the CBMPs are like an out of body experience. I can carefully, almost dispassionately, observe how I am feeling, evaluate the situation in a balanced rather than hyperanxious manner, and think through appropriate courses of action. In the past, I would react first and deal with the consequences later, and, as is common for abuse survivors, would frequently exist in a disassociated state. Since taking CBMPs, I no longer experience extensive periods of disassociation, but instead can be fully present during most interactions and activities. I am able to be mindfully focused on, and completely present, in any situation, without the gnawing anxiety associated with work, other pressures or intrusive thoughts of my abuse, past trauma and the threat of disaster which had previously dominated my internal state and negatively impacted my emotional focus and consistency. I have also managed to break through crushing writer’s block which had been hindering my academic productivity for years. In the following section, I will unpack some of these positive changes and compare them with the lived experiences of other medical cannabis patients.

(i)Managing symptoms:

Talshir et al. [17] conducted interviews with eight Israeli women with PTSD following sexual assault who used cannabis to help manage their symptoms, which included insomnia, night terrors, anxiety and panic attacks. One of Talshir et al.’s interviewees used the analogy of cancer remission. The cannabis could not cure the trauma she had experienced, but it provided the support needed for the trauma to recede into the background. Another of their interviewees, Aviv, stated “It’s been years that I’ve been diagnosed with post-trauma and anxiety disorder, and the cannabis changed my life. After years [during which] I barely slept, I found this wonderful plant. Since then, I’ve been sleeping, my anxiety attacks are down by at least 50%, and in general my life changed from surviving day-by-day to a calm and peaceful life.”, while another, Na’ama, stated “You can speak, you can contain [your] feelings. Suddenly, everything is ok. It’s calmer, and you can come back down to earth.”

These reported experiences share some similarities as well as some differences with my own. As noted above, the CBMPs have also helped me to contain my feelings, but I would describe the medical cannabis as facilitating an out of body experience. As I was usually so closely and emotionally bound up in a situation, this seems qualitatively different from a coming back to Earth. Rather, with the help of the CBMPs, I am able to step back and away from being emotionally overwhelmed and see the situation clearly, in a manner which sometimes prompts me to think of the view that astronauts have of the planet—a zooming out and creation of distance and perspective—which is simultaneously very grounding. The following account is an excerpt from my diary entry documenting my first dose of CBMPs, which elucidates this apparent paradox.

“The drops taste unpleasant and I do not enjoy holding them under my tongue as is the required mode of delivery for these oils. I have to struggle not to spit them out and am reminded why recreational cannabis has never appealed—I simply can’t abide the smell and taste of this plant. The pungency of the medicine exacerbates my bad mood and the panic starts to rise as I question the wisdom of what I am doing. But within less than half an hour I can feel a profound change. I return to my desk and the stressors—the emails, the to do list, the intrusive thoughts, the bills, the chores I have yet to tackle—loom. I start to go through the motions of reacting, but then realise I just don’t feel the usual panic or anxiety. It takes me aback. Another email alert with a problem at work. I skim it and automatically start to compose a response, but my reaction is slightly delayed, and in that split second I find I can exhale. In doing so, the urgency and the panic is gone. I feel mildly sedated, body uncharacteristically still while my mind seemingly expands to accommodate and envelop what would previously have overwhelmed me. There is a pleasant, dreamlike quality to this experience. I find myself able to dispassionately assess the situation from a place of calm, like I have stepped outside of my body and am an external observer. I realise I haven’t broken my pen, which usually happens when I encounter something stressful while at my desk. This too feels oddly emancipatory. I look down to examine my hands which are now just resting in my lap, gently holding the intact biro. Over the many years of therapy, the key take home point had always been to try and step back. To implement some intervention as soon as I felt something triggering an attack, the moment my breathing changed, my heart rate spiked, my stomach lurched, my nails dug into my palms, my pen snapped, or my field of vision and window of tolerance began to narrow. I had never been able to do that. I knew the grounding exercises, the breathing techniques, the power of music or movement, I read (but never believed) the motivational post-it notes my therapist had suggested I stick strategically about the place but in the moment it was always impossible. Now, for what feels like the first time in my life, I understand what it means to step back and more importantly, I feel I finally have the capacity to implement this new understanding.”(Day 1 of Month 1)

The next month, another diary entry provides a more concise reading of the situation that I now find myself in thanks to the CBMPs.

“It’s very strange the way the cannabis enables me to cordon off emotions or feelings and work productively around them—I know they’re there but they’re now like background noise and I’m able to examine them from a distance to work out how best to tackle whatever triggered them.”(Day 26 of Month 2)

The following month, these abridged entries from two consecutive days also reveal the positive impacts of CBMPs, even in the face of objectively high-stress situations.

“crazy day with [partner] being rushed to hospital. Managed to cope and stay calm and do everything that needed to be done both at work and home, and be completely emotionally present and available despite being exhausted and worried. It’s like I’m watching what’s going on through a filter or window, or like I’ve been imbued with superhuman strength of mind.”(Day 17 of Month 3)

“some high stress meetings and emails but felt distant from it, like it wasn’t happening to me, and in that place I was able to work through it all productively and not dwell on anything negative.”(Day 18 of Month 3)

Another entry a month later reveals how the CBMPs are not a panacea, but nonetheless provide much needed relief and protection against panic and anxiety triggers.

“low level anxiety was a constant all day, but [I’m] also on low dose due to running out and delay before next prescription will arrive. Very worried about [partner]. Given the level of worry and how long it [his ill health] has been going on, that the anxiety was low level is testament to the power of the CBMPs. But for the first time since starting the trial I’m existing again in a state of hypervigilance. Awoke in the night after hearing B [dog] making an odd noise. Got up to check him and couldn’t see that he was breathing. He was completely inert even when I started trying to physically rouse him and for an instant it seemed as though he’d died so I cried out. That woke him. He was fine, just really groggy after a very deep sleep. While in the moment it was objectively terrifying, the usual fear and panic I would expect to have experienced when confronted with the [in this case imagined] death of a loved one was coated in something that kept it in check and it felt muted. I was able to get back to sleep again myself quickly after that which was also unusual after a panic attack.”(Day 13 of Month 4)

The closest analogy I have been able to come up with is the feeling that I experienced (or rather did not experience) when the epidural started to take effect late into the difficult labour preceding the birth of my daughter. I had been through several days of extremely painful contractions, but due to overcrowding on the maternity wards and being slow to dilate, I had been sent home twice. When my waters finally broke and I was taken to a birthing room, my plans for a natural process went out of the window and I begged for an epidural, despite chronic trypanophobia (fear of needles). The relief was indescribable. The physical pain dissipated almost immediately, and I became much more aware of what was going on. I could feel some of the mechanics of giving birth, but the anaesthesia freed me to focus on breathwork and pushing.

(ii)Task completion:

Another significant and positive impact of the CBMPs which became apparent very early on in the course of treatment was its seeming ability to enable me to overcome some of the obstacles which had been hampering my professional progress. In my academic career, I have consistently prioritised teaching and the supervision of research postgraduate students at the expense of my own research outputs, due in part to crippling self-doubt or what is termed ‘imposter syndrome’, as well as difficulties with time management and boundary issues which Maté [18] has argued are commonly experienced by individuals with ADD and PTSD. Due to workaholism (an escape or coping mechanism which Maté also links to trauma and ADD), I have been able to be productive over the years, but since the previously buried Pandora’s box of my childhood abuse was unearthed and unlocked during intensive therapy following the birth of my daughter, my ability to complete my own writing projects had pretty much dried up. This led to disappointment, frustration and anxiety on my part, as well as stigma, guilt and shame from feeling that I was letting down colleagues, collaborators, co-authors and publishers, which became a source of deep depression. Much of the extant qualitative research focusing on the experiences of medical cannabis patients has emphasised the stigma that they feel is associated with medical cannabis use itself. Nelson, for example, discussed her own anxiety about external perception, asking “Will others think less of me because I am a medical cannabis patient and advocate?” [19], before describing the complex stigma felt by her family, stating, “My parents fear that others will see me as a less competent, less intelligent woman simply because I use [medical] cannabis.”

Nelson herself acknowledges that “Although I do not always find acceptance as a cannabis patient outside of the medical cannabis community, I become stronger and more resolute each time I share my story. [...] I’m confident that as more patients share our private stories publicly, others will come to have a better understanding of what it means to be a medical cannabis patient—and through this understanding the stigmas associated with cannabis use will dissipate and the use of cannabis as medication will become normalized...” [19]. My perspective is again slightly at odds with the existing prevalent emphasis on stigma as something that medical cannabis patients feel as a result of taking CBMPs. For me, the CBMPs have alleviated the pre-existing stigma and depression associated with writer’s block caused by anxiety, aiding me in ‘doing the work’ (as in the practical labour required to succeed in my profession, as well as the psychological, emotional labour needed for healing) quite literally. During the first four months of taking CBMPs, I was able to complete two long-overdue book chapter contributions, both of which had been very difficult to write due to their emotive subject matter. I have also been able to resume work on several other partially completed manuscripts and collaborative writing projects, and am making significant and productive steps to move them forwards.

(iii)Risk vs. benefits:

It is nonetheless important to acknowledge that it has not all been plain sailing and that CBMPs are not a panacea. Indeed, many extant studies have also identified that there are sometimes negative impacts which need to be weighed against the positive benefits. For Bar-Lev Schleider et al. [7], for example, 5.9% of the young people with ASD in their study reported negative side effects, including increased restlessness. Talshir et al. [17] also found that “the main challenge with cannabis use in the context of PTSD is the feeling of detachment that it engenders.” While this state enabled interviewees to cope with their symptoms, it had the negative outcome of creating a sense of dissociation and an inability to feel “fully present in their relationships and everyday life”, while at the same time increasing “feelings of self-alienation and helplessness.” As the vignettes from my diary above reveal, for me, CBMPs relieved dissociation and enabled me to be fully present in my life in ways that had previously eluded me.

In my experience, there have been some occasions when the dose does not do what I need it to do and where I have had to ‘top up’ (i.e., take an additional dose of CBD) during the day. I have sometimes felt itchy or restless when trying to get to sleep when I increase the dose of THC above 0.2 mL per day, or I have felt drugged and sluggish if I have taken my THC dose too early in the evening, but these are minor inconveniences and overall it has indeed been life-changing. CBMPs have enabled me to regain some semblance of order and to overcome the disabling symptoms of chronic anxiety, PTSD and depression that stem from childhood abuse and other traumas.

Writing about my mental health and the impacts of CBMPs on it raises another area where my own experiences intersect with those of some other medical cannabis patients. Exposing the psychological challenges that I have faced and acknowledging the beneficial outcomes of CBMPs on my wellbeing and interpersonal relationships is not without risk. Nelson, for example, states “When I was raising my family in Oklahoma and Texas, I spent many years fearing that my children might be removed from my home if others discovered I used marijuana. Although I am no longer plagued by this particular fear, I have spoken with many patients—particularly mothers—who are still quite fearful that their use of cannabis as medication will result in a similar fate.” [19]. Nelson is based in the USA where the legality of medical cannabis varies from state to state, and she had not always lived in states where her cannabis use was legal. In the UK, where I am based, the legal use of a prescribed medication to improve mental health does not carry the same risks. Nonetheless, I do feel fear about disclosing so much deeply personal information. This fear is a form of the stigma so widely acknowledged in the extant literature, a fear that people might form harsh judgements about me as an individual, as a professional academic and as a mother because both mental health and medical cannabis are still heavily stigmatised, including within academia. However, as Campbell, who wrote an autoethnography of her own depression and anxiety in the context of an academic workplace, notes, “if we continue to hide uncomfortable stories then we simply perpetuate the taboo or stigma attached” [20]. As a result, she argues for “opening a conversation on difficult matters.” In the context of medical cannabis, Nelson also advocates for the importance of patients making “their private stories public.” [19]. The desire to challenge the existing stigma around these intersecting issues—mental health in academia and medical cannabis—is one of the main motivations for writing about my experiences. Indeed, as Obegu et al. have argued in relation to the evidence base needed to bring medical cannabis into mainstream use, “Centering equity and lived experience strengthens the rationale for investments and ensures user-led evidence generation and utilization—a key public health gain” [21].

There is one further stigma that I hope my story might help to confront. According to the most current statistics at the time of writing, produced by the Office for National Statistics in 2020, 1 in 5 adults in the UK experienced some form of abuse in childhood, while “3.1 million adults aged 18 to 74 years were victims of sexual abuse [specifically] before the age of 16 years” [22]. Yet, despite abused children being victims, childhood abuse is stigmatised. Until now, there have been very few occasions when I have felt comfortable disclosing this aspect of my identity. When I have shared it, it has been difficult for all of those concerned, myself included, to know how to act or react. I feel immense shame about what I endured, and as I have discussed in part here, the effects of the abuse and trauma that I experienced have pervaded every facet of my life. Not every abuse survivor has the opportunity to share their story however, and many will no doubt be suffering in silence, as I did for more than three decades until the desire to protect my unborn child prompted me to finally seek help, if not restitution. Given the number of individuals who have experienced childhood abuse then, the potential for healing provided by CBMPs needs to be explored as a “key public health gain” in addition to an act of social justice.

## 4. Conclusions

While I have only been taking CBMPs for four months at the time of writing, my diary entries show sustained improvements to my mental health and wellbeing. My individual lived experiences of anxiety and CBMPs therefore suggest that CBMPs represent a potentially beneficial intervention for patients whose anxiety has previously been resistant to conventional treatments. As a concluding thought, I am reminded of critical discussions within the social sciences and humanities regarding disability activism. What is termed the social model of disability advocates for the removal of the barriers that disabled individuals face which have been imposed by society [23]. For many disabled individuals, myself included, certain circumstances have created or perpetuate these obstacles. The trauma of experiencing childhood abuse to all intents and purposes disabled me, and despite greater societal and medical awareness of the scale and impacts of abuse, it arguably remains an issue steeped in social stigma. Based on my experience, and those of many of the other medical cannabis patients reviewed in the extant literature, greater access to and acceptance of this plant-based medicine could represent one significant step in removing the obstacles faced by abuse survivors as well as individuals who experience disabling anxiety for other reasons.

## Data Availability

Due to the personal nature of diary entries which form the dataset, data sharing is not applicable to this article.

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
