# Peer review of "An Individual’s Lived Experiences of Taking Cannabis-Based Medicinal Products (CBMPs) to Treat Anxiety"

_ijerph, 2023, doi:10.3390/ijerph20186776_

Round 1
Reviewer 1 Report
This is a novel case report with the potential to contribute to broader understanding of the benefits of cannabis for patients suffering from serious mental health conditions. As a sociologist with no previous experience reading or writing case reports, I make the following suggestions for revisions that would benefit the manuscript in my view.
Since the case report submitted is actually, and correctly, identified as being an autoethnography, I recommend expanding on the very brief description of the methodology with a more robust discussion of other contributions to the literature, like this one, that inform the authors autoethnographic approach. This might include non-academic sources of non-fiction with depictions of those suffering from mental health conditions and stories of recovery from a first-hand point of view.
Similarly overlooked is an extensive literature of narratives of persons who use cannabis for coping with mental health conditions. Since this literature exists, it is important to review it to better situate this work within similar discussions and make its contribution to the knowledge base clearer.
Although the title would suggest that this is a case report on an individual’s experience with cannabis, it is really more about the mental health condition, including harrowing accounts of several episodes and symptoms. These descriptions call forth empathy, concern, and some relief that cannabis is helpful as a therapeutic tool. Nonetheless, other readers like myself would benefit from learning something more about the actual experience with cannabis in terms of how it helps, and how it does not help or otherwise, and how these experiences square with, reflect or otherwise, the accounts of others that we have read about.
Author Response
Thank-you for giving me the opportunity to revise and resubmit this paper. The reviewers’ comments were constructive and below I have set out how I have addressed them. I have used track changes on the revised document.
- As requested by reviewer 1 I have provided a more robust discussion of methodology and incorporated an expanded discussion of other contributions to the literature. However, I have kept these additions limited to academic sources due to word count restraints.
- As requested by reviewer 1 I have also added additional narratives from persons who use cannabis for mental health conditions.
- As requested by reviewers 1 I have now provided a detailed account of how the medical cannabis has positively impacted on my symptoms so there is balanced coverage of before and after.
Reviewer 2 Report
The author shared a very interesting personal experience of how medical cannabis helped her cope with anxiety and other conditions (possibly related to PTSD?).
1) A timeline or a chart would be useful to illustrate the changes in the author's symptoms, stress levels, and functioning before and after using CBMPs.
2) The description of the author's experience before using CBMPs is much more detailed than the one after using CBMPs. A more balanced presentation would make the comparison easier.
3) Page 3, "I offer suffer relapses..." should be corrected to "I often suffer relapses..."
4) The author should also mention if she was undergoing any other therapy or medication before and after CBMPs.
Author Response
Thank-you for giving me the opportunity to revise and resubmit this paper. The reviewers’ comments were constructive and below I have set out how I have addressed them. I have used track changes on the revised document.
- As requested by reviewer 2 I have now provided a detailed account of how the medical cannabis has positively impacted on my symptoms so there is balanced coverage of before and after.
- Reviewer 2 requested a timeline or chart be included. However, this did not sit comfortably with the descriptive, qualitative approach of autoethnography. As a compromise, a series of vignettes from across the focal timeframe were included and dated to demonstrate the progression.
- The typo identified by reviewer 2 has been corrected.
- As per reviewer 2’s final request, I have clarified what treatment I had received prior to taking CBMPs and confirmed I have not undergone any other treatment since starting on the trial.
Round 2
Reviewer 1 Report
The author has effectively responded to my previous suggestions comprehensively. I have no further concerns and recommend for publication in its present form.